# Development of an Instantaneous Loading Impact Test System for Containment of a Nuclear Power Plant during Aircraft Impact on Steel Bar Joints

**DOI:** 10.3390/ma16103892

**Published:** 2023-05-22

**Authors:** Wanxu Zhu, Shiyuan Liang, Kefei Jia, Quanxi Shen, Dongwen Wu

**Affiliations:** 1Guangxi Engineering Research Center of Intelligent Structural Material, Guilin University of Technology, Guilin 541004, China; 2Guangxi Key Laboratory of Geotechnical Mechanics and Engineering, Guilin University of Technology, Guilin 541004, China; 3Jinan Lizhi Test System Co., Ltd., Jinan 250021, China

**Keywords:** nuclear power station, aircraft impact, reinforcement joint, instantaneous loading, test system

## Abstract

As major projects such as nuclear power plants continuously increase, it is inevitable that loopholes will arise in safety precautions. Airplane anchoring structures, comprising steel joints and acting as a key component of such a major project, directly affect the safety of the project due to their resistance to the instant impact of an airplane. Existing impact testing machines have the limitations of being unable to balance impact velocity and impact force, as well as having inadequate control of impact velocity; they cannot meet the requirements of impact testing for steel mechanical connections in nuclear power plants. This paper discusses the hydraulic-based principle of the impact test system, adopts the hydraulic control mode, and uses the accumulator as the power source to develop an instant loading test system suitable for the entire series of steel joints and small-scale cable impact tests. The system is equipped with a 2000 kN static-pressure-supported high-speed servo linear actuator, a 2 × 22 kW oil pump motor group, a 2.2 kW high-pressure oil pump motor group, and a 9000 L/min nitrogen-charging accumulator group, which can test the impact of large-tonnage instant tensile loading. The maximum impact force of the system is 2000 kN, and the maximum impact rate is 1.5 m/s. Through the impact testing of mechanical connecting components using the developed impact test system, it was found that the strain rate of the specimen before failure was not less than 1 s^−1^, meeting the requirements of the technical specifications for nuclear power plants. By adjusting the working pressure of the accumulator group, the impact rate could be controlled effectively, thus providing a strong experimental platform for research in the field of engineering for preventing emergencies.

## 1. Introduction

With the continuous increase in major construction projects such as nuclear power plants, safety measures are inevitably becoming prone to loopholes. The risks caused by impacts to nuclear power plants, such as nuclear leakage, have attracted considerable worldwide attention, and the impact resistance requirements of nuclear power plant containment structures have become increasingly stricter [1]. During sudden disasters, the steel reinforcement joint of the safety shell in a nuclear power plant often bears large impact loads [2]. Although the material has good plasticity under static loads, it may exhibit brittleness under impact loads with a strain rate exceeding 1 s^−1^ [3]. Therefore, the impact resistance of steel reinforcement joints directly affects the safety of engineering during sudden disasters [4]. In 2016, the National Nuclear Safety Administration of China issued a new version of Nuclear Power Plant Design Safety Regulations (HAF102-2016) [5]. Due to the lack of large-tonnage high-speed impact test equipment, research in China on the impact performance of engineering connecting structures is basically nonexistent, and the related products are in the stage of imitation only. Therefore, the lack of a large-tonnage instantaneous loading impact test system has become a key bottleneck problem that hinders the research on the impact resistance of reinforcement joints and other connecting anchorage structures, restricting the development of anti-impact connection anchorage products and affecting the construction of major infrastructure such as nuclear power plants.

Europe, America, Japan, and other countries lead the research on impact testing machines and have developed various forms of them, such as free-fall hammers [6], pendulum hammers [7], vibration impact [8], pulse-type [9], explosion [10], and high-speed tension [11], which are widely used in material research [12,13,14,15,16,17]. Agirre et al. [18] constructed an automatized forging hammer that is accelerated with a pneumatic cylinder and is able to speed up the upper anvil up to 5 m/s. However, the maximum testing force is only 250 kN. The 1000 kN high-speed impact testing machine in Germany’s BAM laboratory can perform the instantaneous tensile impact testing of a specimen at a strain rate of 1 m/s and maintain a constant rate in the yield stage; this is currently the only equipment in the world qualified for the instantaneous loading impact testing of steel reinforcement joints for resisting aircraft collision.

In recent years, China has gradually made significant progress in the research and development of impact testing machines. Pendulum impact [19] and metal drop hammer testing machines [20] are currently widely used. In the latter device, the magnitude of the impact load is determined by the mass and height of the hammer. The highest impact speed of the machine is 19.8 m/s, which facilitates fast impact testing. However, the maximum mass of its hammer is only 490 kg and the impact force is also small, which cannot meet the requirements of impact testing for connection anchorage structures [21]. For specific engineering applications and structural components, hydraulic impact testing machines that can achieve a large-tonnage impact force have also been developed in China. Wang et al. [22] proposed a hydraulic-driven heavy-duty anti-impact test system model for ship equipment with a maximum impact velocity of 5 m/s, but its impact force still needs to be improved. Xie [23] and Xu [24] designed high-pressure and large-flow safety valve test devices powered by accumulators, whose testing confirmed that the loading system powered by accumulators can quickly achieve the target impact pressure required for the test. In 2018, a fast hydraulic impact test platform was designed at Liaoning Technical University [25] for low-strain-rate impact tensile tests of mine anchor rods; it meets the performance requirements of a 500 kN impact force and impact velocity up to 6 m/s, but the impact force cannot yet meet the requirements of a Φ32 threaded rebar joint. In 2020, Zhang et al. [26] designed a hydraulic energy storage and fast-release-principle-based impact testing machine for supporting impact-resistant rock strata in mines with a maximum impact force of 6500 kN and a maximum impact velocity of 8.2 m/s. Although it can only perform unidirectional compression impact tests on specimens, it further verifies the feasibility of adopting the hydraulic method and an accumulator as the power source for large-scale impact systems.

In summary, the existing impact testing machines have the limitations of being unable to balance impact velocity and impact force, as well as having inadequate control of impact velocity; they cannot meet the requirements of impact testing for steel mechanical connections in nuclear power plants. This study aims to develop an instant loading testing system for the entire series of steel joints used in the safety shell of nuclear power plants, which would be able to withstand airplane crashes, using a hydraulic control method and a spring accumulator as the power source. By adjusting the working pressure of the accumulator group, the impact velocity can be controlled effectively, thus providing a powerful testing platform for the field of engineering disaster prevention and mitigation.

## 2. Hydraulic Principle of Instantaneous Loading Impact Test System

### 2.1. Working Principle of Valve-Controlled Hydraulic Cylinder of Impact Test System

This experimental system is a high-flow instantaneous loading impact test system, which uses a valve-controlled symmetric hydraulic cylinder. The schematic diagram of the system is shown in Figure 1.

In the power mechanism of the valve-controlled hydraulic cylinder, the load pressure PL and load flow rate Q2 are two fundamental quantities. The valve-controlled symmetric hydraulic cylinder mechanism is usually defined as follows:(1)$$\left\{\begin{array}{l} P_{L}=P_{1}-P_{2}\\ Q_{L}=(Q_{1}+Q_{2})/2 \end{array}\right.$$
where *P_L_* is the load pressure, *Q_L_* is the load flow rate, and *Q*_1_ and *Q*_2_ are the flow rates into and out of the hydraulic cylinder chamber, respectively.

Since the hydraulic cylinder is a symmetric cylinder, the flow rates into and out of its chamber are equal. Therefore:(2)$$Q_{L}=Q_{1}=Q_{2}=\frac{\left(Q_{1}+Q_{2}\right)}{2}$$

The hydraulic power input to the hydraulic cylinder, which is also the hydraulic power output of the valve, can be obtained as:(3)$$N_{L}=P_{L}Q_{L}=\frac{\left(P_{1}-P_{2})(Q_{1}+Q_{2}\right)}{2}$$

As shown in Figure 1, the load in the hydraulic cylinder can be calculated as follows:(4)$$F_{L}=A(P_{1}-P_{2})$$

The speed of the hydraulic cylinder can be obtained as:(5)$$V=\frac{Q_{1}}{A}=\frac{Q_{L}}{A}$$

Then, the output mechanical power of the hydraulic cylinder will be:(6)$$N_{C}=F_{L}V=\frac{A(P_{1}-P_{2})Q_{L}}{A}=P_{L}Q_{L}=N_{L}$$

That is, for the valve-controlled symmetrical hydraulic cylinder mechanism under ideal conditions, the hydraulic power output by the oil valve is equal to the mechanical power output by the hydraulic cylinder.

### 2.2. Selection of Hydraulic Accumulator for Impact Test System

The developed impact testing system uses a bladder-type accumulator, which has a good pressure response, a large capacity, high pressure, and a wide temperature range. The working principle of the accumulator is shown in Figure 2. When the oil valve is closed, the bladder is precharged with nitrogen gas, with *P*_0_ as the precharge pressure and *V*_0_ as the effective gas volume. When the accumulator is at its minimum working pressure, a small amount of oil should be retained between the bladder and oil valve to prevent the bladder from being damaged by the hitting of the oil valve during each expansion process. At this stage, *P*_1_ is the minimum working pressure and *V*_1_ is the corresponding gas volume. When the accumulator is at its maximum working pressure *P*_2_, *V*_2_ is the corresponding gas volume. The change in volume between *V*_1_ and *V*_2_ is equivalent to the effective volume.

The proportion between the nitrogen precharge pressure *P*_0_ and the maximum operating pressure *P*_2_ of the accumulator is limited to 1:4. The precharge pressure should not exceed 90% of the minimum system pressure *P*_1_, thus limiting the range of deformation of the bladder during operation and prolonging its service life.

According to Boyle’s law:(7)$$P_{0}V_{0}^{n}=P_{1}V_{1}^{n}=P_{2}V_{2}^{n}=C$$

The change in volume of the single tank accumulator can be expressed as:(8)$$\Delta V=V_{1}-V_{2}=V_{0}P_{0}{}^{\frac{1}{n}}[{\left(\frac{1}{P_{1}}\right)}^{\frac{1}{n}}-{\left(\frac{1}{P_{2}}\right)}^{\frac{1}{n}}]$$

The change in volume of the accumulator is approximately equal to the oil output. If the total number of open accumulator tanks is *N*, then the total oil output can be obtained as:(9)$$Q=\Delta VN=V_{0}P_{0}{}^{\frac{1}{n}}[{\left(\frac{1}{P_{1}}\right)}^{\frac{1}{n}}-{\left(\frac{1}{P_{2}}\right)}^{\frac{1}{n}}]N$$

For an adiabatic process with n equal to 1.4, Equation (9) becomes:(10)$$Q=V_{0}P_{0}{}^{0.715}[{\left(\frac{1}{P_{1}}\right)}^{0.715}-{\left(\frac{1}{P_{2}}\right)}^{0.715}]N$$

To ensure that the specimen can be pulled apart, the total oil output *Q* should satisfy:(11)$$Q\geq Q_{0}=x_{\max}\Sigma A$$

Meanwhile, the supplied oil pressure *P* should satisfy:(12)$$P\geq \frac{F_{b}}{\Sigma A}$$

In Equation (12), $x_{\max}$ represents the ultimate elongation of the steel bar joint, ΣA is the total hydraulic pressure area of the pipeline, and $F_{b}$ is the maximum force sustained during the tensile test.

The impact testing system uses high-pressure accumulators to supply oil and converts hydraulic energy into impact energy through a valve-controlled cylinder system. Its dynamic characteristics are complex, and impact velocity is closely related to the parameters of the accumulator. In order to ensure that the specimen is properly fractured, as per Equation (11), the total oil output should not be less than the product of the ultimate elongation of the steel bar and total hydraulic pressure area of the pipeline. As seen in Equation (10), when the number of cans opened in the accumulator is fixed, the desired oil output can be obtained by adjusting the energy storage pressure. By investigating the effect of different energy storage pressures, the impact rate can be controlled effectively.

## 3. Composition of Transient Loading Impact Test System

### 3.1. Structure of Instantaneous Loading Impact Test System

The electrohydraulic instant load impact testing system is composed of a large-space four-column host frame, a static pressure support high-speed servo linear actuator, a hydraulic servo pump station, a high-pressure forced clamp head, a control software system, etc. It is suitable for conventional static mechanics performance tests of engineering materials and components, high-speed impact tensile tests, high-speed compression tests, high-speed penetration tests, etc. It can also perform one-way tensile tests of steel joints, high-stress repeated tensile tests, and large-deformation repeated tensile tests. It integrates the functions of electrohydraulic servo automatic control, automatic measurement, and data processing. The schematic diagram and physical diagram of the impact testing system are shown in Figure 3 and Figure 4, respectively.

The mainframe of the four-column host consists of the loading host structure, moving crossbeam, worktable base, lifting platform, and electrical control system. The adjustment of the test space involves the symmetric arrangement of the large stroke plunger-type engineering lifting platform, which can be continuously adjusted within the adjustable range. The servo actuator adopts a ±2000 KN static-pressure-supported high-speed linear servo actuator, which is specially designed for high-speed loading with low damping and a high-response structure to ensure that the high-speed performance of the actuating cylinder is stable, reliable, and has a good sealing effect and long service life. The forced hydraulic clamp is composed of a clamp and a hydraulic drive module, which clamp the test piece. The clamp is composed of a fixture body, a clamping piston, pliers, and related accessories. The clamping cylinder is embedded in the fixture body, and it adopts a two-way clamping method for ensuring the coaxiality of the specimen regardless of the change in thickness. The load sensor is a ±2000 kN shear-strain-type load sensor with an overload capacity of 150%, while the displacement sensor is a large-range magnetostrictive displacement sensor with noncontact measurement, infinite mechanical life, and high linear accuracy, where the magnetic block measurement is independent of the moving speed. The hydraulic pump station and nitrogen-charged accumulator group consist of a main oil tank, an oil pump motor group, a high-pressure oil pump motor group, a plunger pump, a high-pressure plunger pump, a superimposed valve, a reversing valve, and a control valve block. The pump station adopts the design of an expandable structure, which can be conveniently upgraded and transformed into subsequent test functions. The nitrogen-charged accumulator group is equipped with a dedicated nitrogen-filling trolley, which is convenient for replenishing nitrogen. The hydraulic pump station is equipped with a self-contained air-cooling system to ensure reliable and stable operation during the testing process. The measurement and control system uses a fully digital controller and industrial control computer software system. The user-friendly interface can display important test parameters and information in real time, making it easy for operators to monitor the working condition of the testing system at any time. The hydraulic pipeline system is configured according to the layout of the testing laboratory. A pressure regulator and a distributor are installed near the main unit. The pipeline from the pump station and nitrogen accumulator unit to the pressure regulator and distributor is made of a hard metal, while the pipeline from the pressure regulator and distributor to the main unit is made of high-pressure hose assemblies. These components form the instant loading and impact testing system. According to the Nuclear Power Plant Design Safety Regulations (HAF102-2016), the strain rate of large-diameter steel mechanical joints for impact resistance testing should meet the requirement of 1 s^−1^. Therefore, the maximum tensile impact test force of the test system is designed to be 2000 kN, and the maximum loading rate is 1.5 m/s. The main technical specifications are listed in Table 1.

To address the problem of the unstable clamping of specimens during instant loading, a large-force hydraulic clamp that can withstand high-speed impact loads was developed. Its main components are the fixture body and clamping cylinder; the clamping cylinder is designed to be embedded in the fixture body with a smaller clearance for a longer service life. The clamp uses a bidirectional clamping method to ensure the coaxiality of the specimen. The fixture body has a closed design, and the specimen is held by moving the piston position when clamping it. The hydraulic clamp’s drive module uses a gas–liquid booster pump, and the clamping force can be adjusted to prevent damage to the specimen caused by excessive clamping force during low-load testing while ensuring the reliable clamping of large specimens. The working schematic diagram of the large hydraulic gripping tool is shown in Figure 5.

The control of the instantaneous loading impact test system requires supporting research and development of a measurement and control system that can acquire data with high speed, stability, and accuracy. The measurement and control system of the testing machine is composed of sensors, controllers, control software, and an industrial host. The force sensor adopts a shear-strain-type load sensor with a self-locking nut, which has high resistance to impact, reliable measurement accuracy, and good stability. The actuator cylinder is equipped with a built-in magnetostriction displacement sensor for noncontact measurement, with an infinite mechanical service life and measurement unaffected by movement speed. The linear accuracy is up to 0.04%. The high-performance EDC controller adopts a 133 MHz AMD520 CPU as the main processor, with a minimum control loop time of 1 ms, a sampling frequency of 1000 Hz, and a test force resolution of ±180,000 codes. It has high integration, stability, reliability, and easy operation and adjustment. It uses LZCL_SOFT2.0 professional control software, which can automatically and accurately collect test data according to user requirements within a specified time during the test process. The front and back panels of the EDC controller are shown in Figure 6, and Figure 7 is a schematic diagram of the main interface of the LZCL_SOFT2.0 operating software.

### 3.2. Hydraulic System of Instant Load Impact Testing System

Energy is provided to the instant load impact testing system by the hydraulic loading system. The high-pressure and high-flow liquid of the hydraulic system drives the impact cylinder with a certain impact speed of the cylinder’s piston rod and applies the impact force for the dynamic impact test of the specimen. According to the test requirement of the highest impact force of 2000 kN, the hydraulic loading system of the impact testing machine was designed, whose schematic diagram is shown in Figure 8. The hydraulic system of this test device mainly consists of a 9000 L/min nitrogen accumulator group, a hydraulic oil pump group, a quick-stop control and lubrication group, and a 2000 kN static-pressure-supported high-speed servo linear actuator.

The accumulator group mainly provides high-pressure and ultra-large flow liquid for the actuator of the impact test system to meet the requirements of the instantaneous release of large energy. The nitrogen accumulator group consists of an accumulator connection valve block, a bracket, a connection pipeline, a valve, an accumulator, etc. It is equipped with 20 63-L accumulators, where the instantaneous maximum flow rate can reach 9000 L/min during the shock and tension. By changing the accumulator pressure and number of open accumulator tanks, different impact loads and impact rates can be achieved; the Ishikawa diagram of factors affecting the performance of the system is shown in Figure 9.

The hydraulic oil pump group includes a charging oil pump, a jaw-clamping oil pump, a beam-locking oil pump, and a beam-lifting oil pump. Two 22 kW oil pump motors and 2.2 kW high-pressure oil pump motors, as well as PK1.93L high-pressure plunger pumps, are selected. The accumulator group and oil source part are the power source of the impact. The accumulator group provides large-flow and high-pressure oil, which reduces the requirements for the oil pump displacement.

The quick cut-off control valve group includes an inserting valve and a pressure-regulating mechanism. The pressure-regulating mechanism includes a control unit and high-pressure and low-pressure overflow valves connected in parallel. When the impact reaches the specified stroke, the control unit switches on the high-pressure overflow valve and blocks the insertion valve, so that the oil no longer flows and the buffering brake is completed.

The actuator is a high-tonnage, static-pressure-supported, dual-rod, high-speed servo linear actuator, with a built-in magnetostrictive displacement sensor installed in the actuating cylinder. The “O-shaped” magnetic block is fixed for guidance. Its measurement is not limited by the movement speed, with a linear accuracy of up to 0.04%. The actuator applies static-pressure-supported technology, which uses the static pressure of the hydraulic oil to support the piston rod, suspending it and ensuring the elimination of the bias load effect of the long-stroke reciprocating linear motion. An end buffering system is used to further alleviate the impact of fracture shock on the hydraulic system and the host.

## 4. Test Verification

### 4.1. Verification of Impact Test

The instant loading impact testing system was calibrated at the Shandong Institute of Metrology in China by using a standard device of a 0.3-grade force gauge and 1.0–10 m/s accelerometer. The calibration results show that the maximum test force in both tension and compression is 2000 kN, with the relative error in the indication within ±0.7% and the displacement error within 0.3%. The impact loading rate is ≥1.5 m/s, and all the indicators meet the national metrological standard defined in “JJG139-2014 Tensile, Pressure and Universal Testing Machine Calibration Regulations”.

A verification test on the impact performance was conducted for the full series of extruded standard mechanical splices with diameters of 16 mm, 20 mm, 25 mm, 32 mm, and 40 mm. The test samples were extruded steel bar splices used for anti-aircraft collision, whose structure is shown in Figure 10. The steel bar was made of HRB500 hot-rolled ribbed steel with a yield strength of ≥500 MPa and tensile strength of ≥680 MPa. The materials of the extruded sleeve and connecting screw were 40Cr alloy steel with a yield strength of ≥785 MPa and tensile strength of ≥980 MPa. Taking a steel bar mechanical connection splice test sample of 32 mm diameter and free section length of 1 m as an example, the impact test was performed with six energy storage tanks. The failure mode of the specimen is shown in Figure 11.

It can be seen from Figure 11 that the failure mode of the tested specimens under impact is the tensile fracture at the position of the steel reinforcement, which meets the requirements for engineering use. The force–time curve of the impact test is shown in Figure 12, and the impact velocity–time curve is shown in Figure 13. From the force–time curve recorded by the measurement and control system, it can be seen that the steel reinforcement underwent four stages under impact, namely elastic deformation, yielding, strengthening, and necking, which are similar to those under the unidirectional tensile static load test. From the impact velocity curve, it can be seen that the strain rate of the specimen continued to increase rapidly during the loading process, and the impact velocity at the moment of the specimen failure exceeded 1 m/s. Since the free length of the specimen was 1 m, its strain rate was equal to its loading rate. As shown in Figure 13, the strain rate of the specimen was not less than 1 s^−1^ before it failed, which meets the strain rate requirement of 1 s^−1^ (tolerance ± 20%) in the impact resistance test of large-diameter steel reinforcement mechanical connections for aircraft impact on the containment vessels in nuclear power plants. During the test, at the moment of the sudden fracture of the specimen, the piston of the oil cylinder decelerated and stopped according to a predetermined specific braking method, reducing the hydraulic impact generated at that moment.

The obtained test results show that this testing system can meet the experimental requirements for the instantaneous loading of the entire series of steel reinforcement joints, and it can be used for the impact testing of steel reinforcement joints in China’s nuclear power projects.

### 4.2. Experimental Study on Impact Rate Control

The test sample is a HRB400 hot-rolled ribbed steel bar with a diameter of Φ25. Its total oil demand and minimum oil pressure required for punching were calculated by using Equations (11) and (12), respectively, and the obtained results are presented in Table 2.

As per Equation (10), there is a corresponding functional relationship between the effective gas volume, precharge pressure, minimum operating pressure, maximum operating pressure, number of accumulator tank openings, and total oil output. The following experiment is designed to investigate the change in the impact velocity curve when the maximum operating pressure *P*_2_ is the variable. The experimental parameters are shown in Table 3. Thus, the feasibility of controlling the impact velocity of the impact testing system by adjusting the energy storage pressure under the condition of a constant number of accumulator tank openings is verified. The impact velocity curve with the energy storage pressure as the variable is shown in Figure 14.

From Figure 10, it can be seen that the time taken for the steel bar to break from loading was about 0.25 s, and the strain rate of the specimen increased rapidly. When the minimum working pressure was maintained at 10 MPa, adjusting the maximum working pressure to 12 MPa resulted in a pressure difference of 2 MPa between the maximum and minimum working pressures, and the maximum impact velocity was 0.93 m/s. Adjusting the maximum working pressure to 15 MPa resulted in a pressure difference of 5 MPa, and the maximum impact velocity was 1.04 m/s. Adjusting the maximum working pressure to 18 MPa resulted in a pressure difference of 8 MPa, and the maximum impact velocity was 1.22 m/s. With the increase in the working pressure differential of the accumulator, the maximum impact velocity of the specimen increased accordingly. Therefore, by adjusting the working pressure of the accumulator tank, the impact velocity of the impact testing system can be controlled.

Different specimens require different loads to break, which can be adjusted by increasing or decreasing the number of accumulator openings. The larger the load required for specimen fracture, the greater the total number N of accumulator openings required. Taking the three most commonly used steel bars of Φ25, Φ32, and Φ40 in nuclear power plant containment as an example, the accumulator pressure and number required for different loads and velocities are shown in Table 4.

## 5. Conclusions

This article proposes a design method for a high-tonnage instant loading impact testing system with a maximum impact force of 2000 kN and maximum impact velocity of 1.5 m/s. The hydraulic working principle of the impact system is explained, and the overall structure and composition of the hydraulic system of the testing system are introduced. By conducting functional testing and impact velocity control testing, the following conclusions are drawn:

(1)The impact testing system converts the hydraulic energy into impact energy through a valve-controlled cylinder system. In order to ensure the normal fracture of the test piece, the total oil output should not be less than the product of the ultimate elongation of the steel bar and total hydraulic pressure area of the pipeline. The relationship between the total oil output, accumulator pressure, and number of openings is $Q=V_{0}P_{0}{}^{0.715}[{\left(\frac{1}{P_{1}}\right)}^{0.715}-{\left(\frac{1}{P_{2}}\right)}^{0.715}]N$ By investigating the influence of different accumulator pressures, the impact velocity can be controlled effectively.(2)The hydraulic loading system is designed to mainly consist of a 9000 L/min nitrogen-charged accumulator group, a hydraulic oil pump group, a fast break control and lubrication group, and a 2000 kN static-pressure-supported high-speed servo linear actuator group. The high-pressure and high-flow hydraulic oil drives the impact oil cylinder, which applies the impact force on the tested object for conducting its dynamic impact testing.(3)The relative error in the testing force indication value of the impact testing system is controlled within ±0.7%, and the displacement error is controlled within 0.3%, where all the indicators meet the national metrological benchmark of “JJG139-2014 Tensile, Pressure and Universal Testing Machine Verification Regulations”. The system can effectively help realize the impact testing of the entire series of steel joints. The strain rate of the steel bar before the specimen is broken is not less than 1 s^−1^, which meets the requirements of the “Nuclear Power Plant Design Safety Regulations” (HAF102-2016).(4)When the minimum working pressure of the accumulator remains constant at 10 MPa, and the maximum working pressure is sequentially set to 12 MPa, 15 MPa, and 18 MPa, then the corresponding maximum impact velocities become 0.93 m/s, 1.04 m/s, and 1.22 m/s, respectively. The maximum impact velocity increases with the increase in the difference in the working pressures. By adjusting the pressure and number of the accumulators, the equipment can achieve different loads and velocities. Further research is needed on how to achieve closed-loop control of the equipment to ensure that the impact velocity is continuously and controllably adjustable.

## Figures and Tables

**Figure 1 materials-16-03892-f001:**
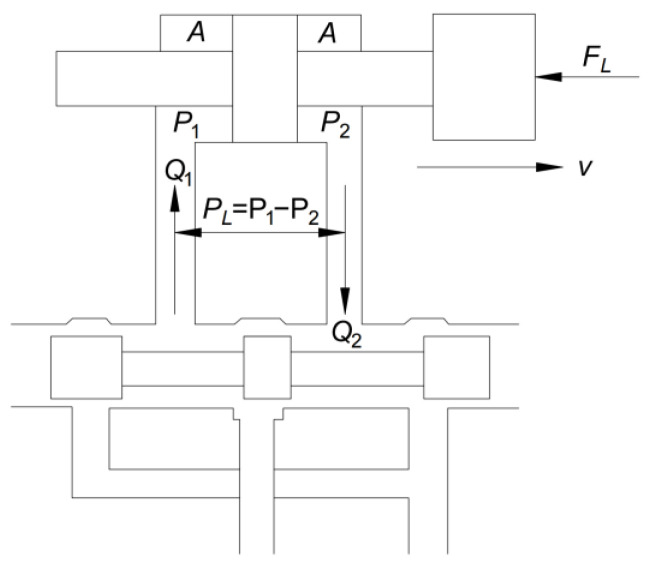
Schematic diagram of valve-controlled symmetric hydraulic cylinder system.

**Figure 2 materials-16-03892-f002:**
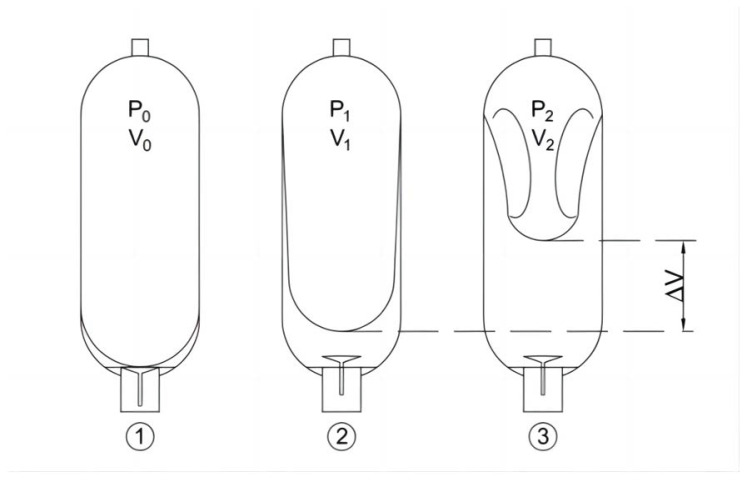
Schematic diagram of working principle of accumulator.

**Figure 3 materials-16-03892-f003:**
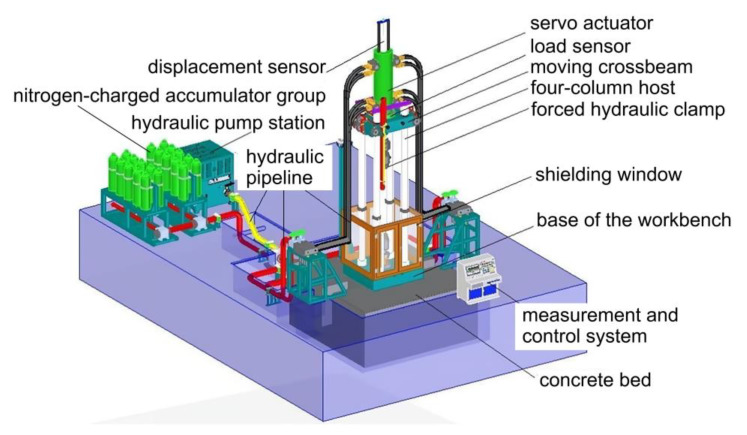
Schematic diagram of transient loading impact test system.

**Figure 4 materials-16-03892-f004:**
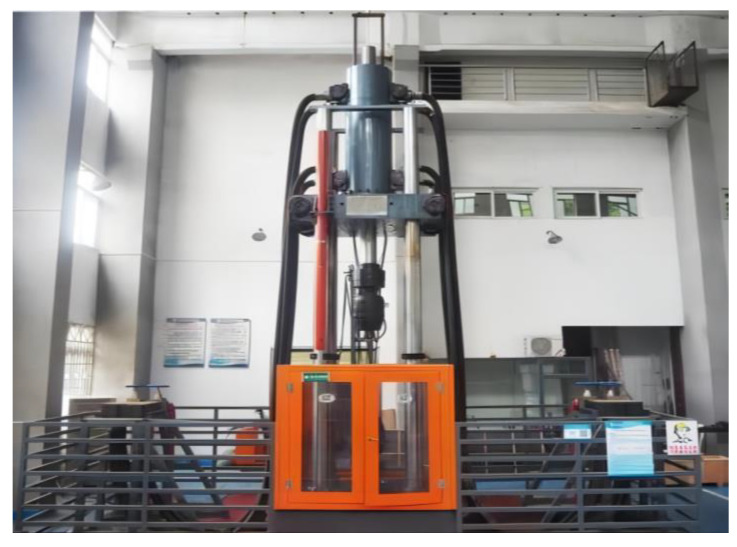
Transient loading impact test system.

**Figure 5 materials-16-03892-f005:**
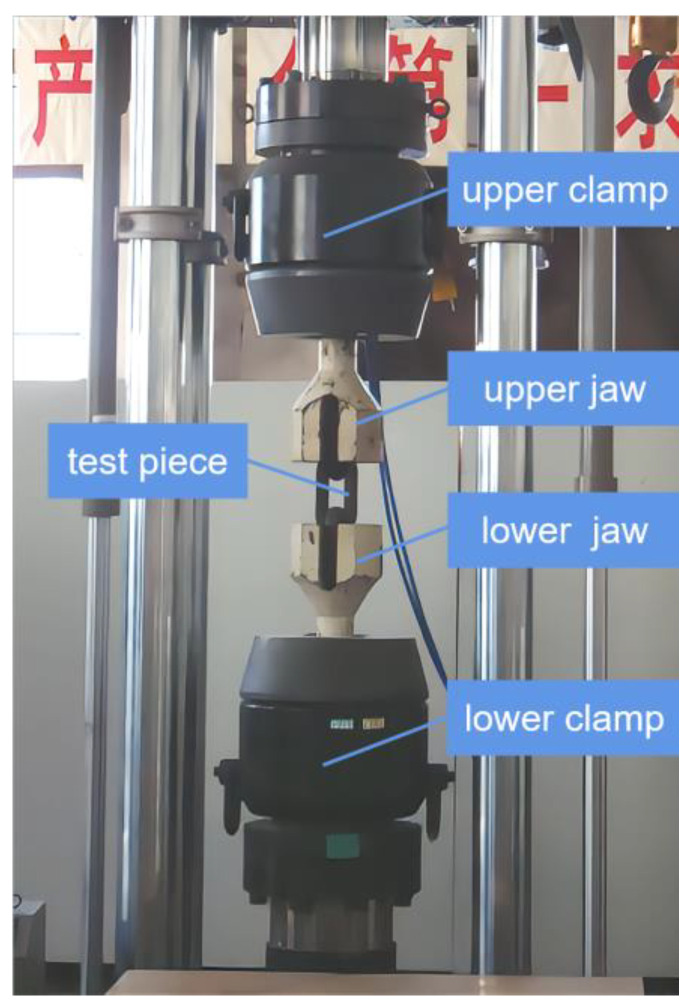
Working diagram of a large-force hydraulic clamp.

**Figure 6 materials-16-03892-f006:**
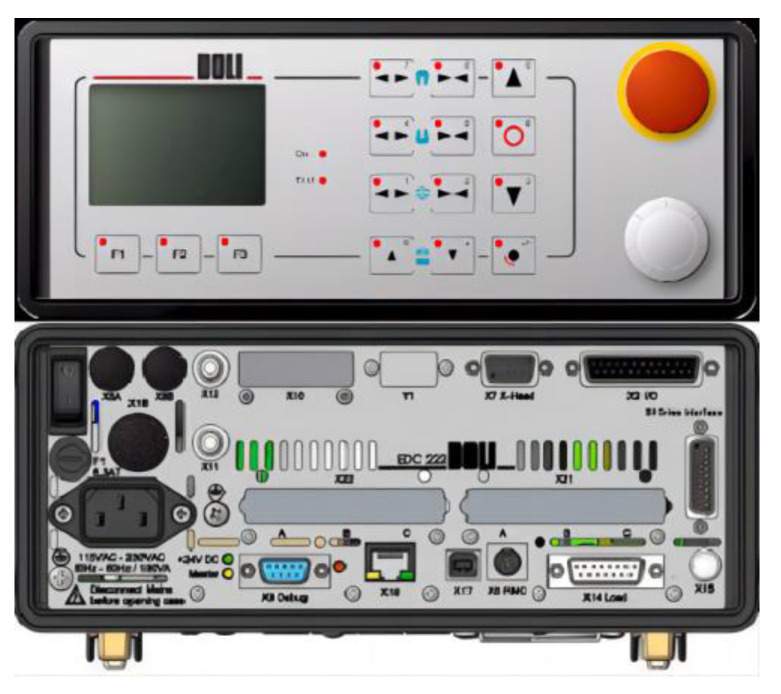
The front and back panels of the EDC controller.

**Figure 7 materials-16-03892-f007:**
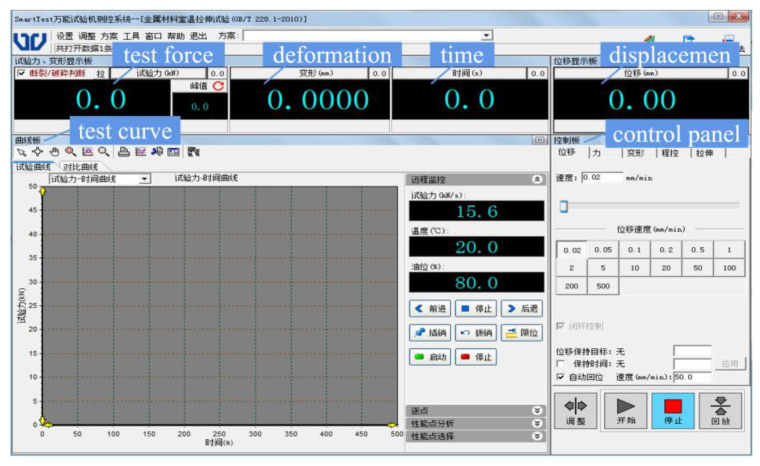
Schematic diagram of the main interface of the LZCL_SOFT2.0 operating software.

**Figure 8 materials-16-03892-f008:**
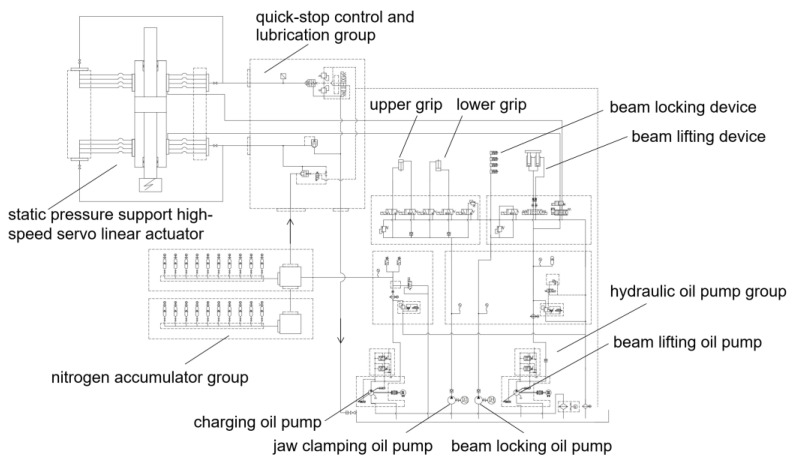
Hydraulic principle of impact test system.

**Figure 9 materials-16-03892-f009:**
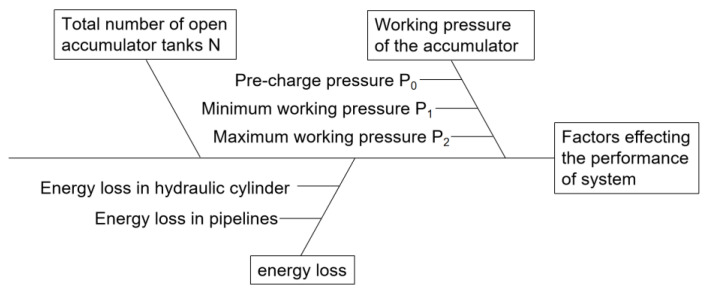
Ishikawa diagram of factors affecting the performance of system.

**Figure 10 materials-16-03892-f010:**
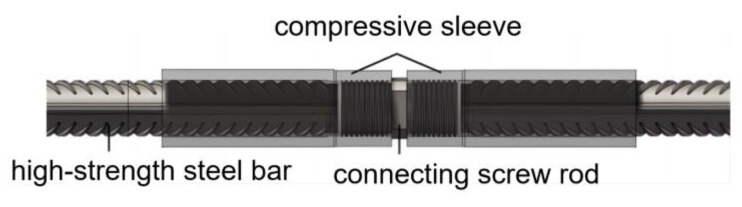
Schematic diagram of mechanical connection of steel reinforcement joints.

**Figure 11 materials-16-03892-f011:**
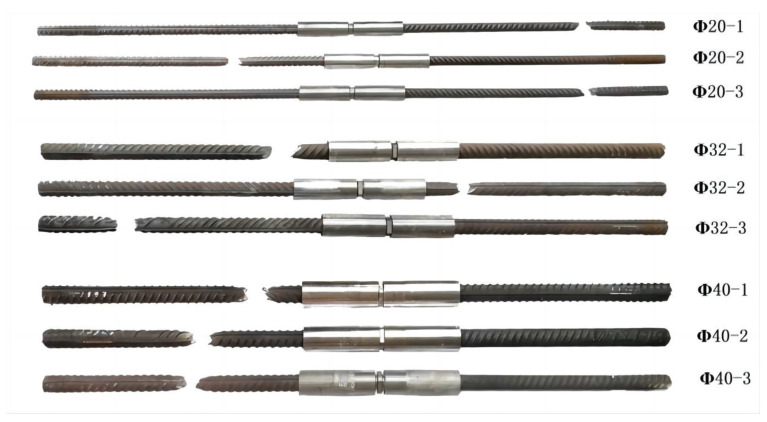
Failure form of test sample.

**Figure 12 materials-16-03892-f012:**
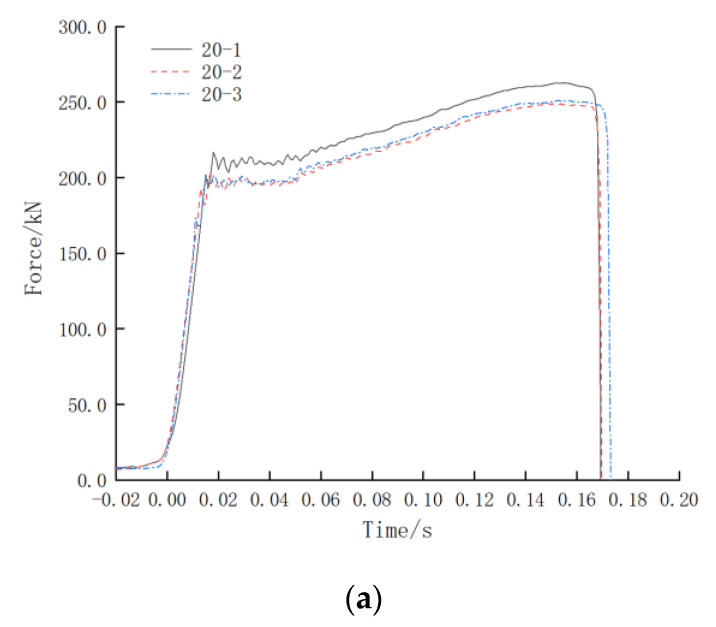
Test force–time curve: (**a**) Test force–time curve of Φ20; (**b**) Test force–time curve of Φ32; (**c**) Test force–time curve of Φ40.

**Figure 13 materials-16-03892-f013:**
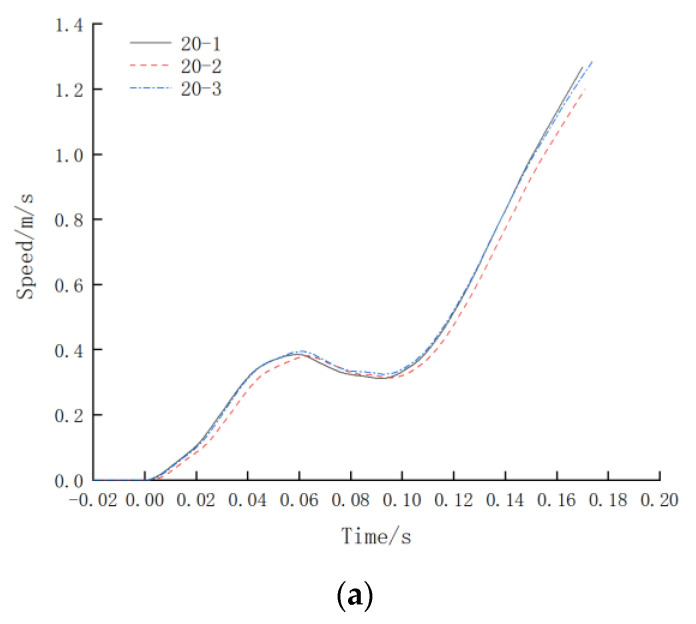
Impact velocity–time curve: (**a**) Impact velocity–time curve of Φ20; (**b**) Impact velocity–time curve of Φ32; (**c**) Impact velocity–time curve of Φ40.

**Figure 14 materials-16-03892-f014:**
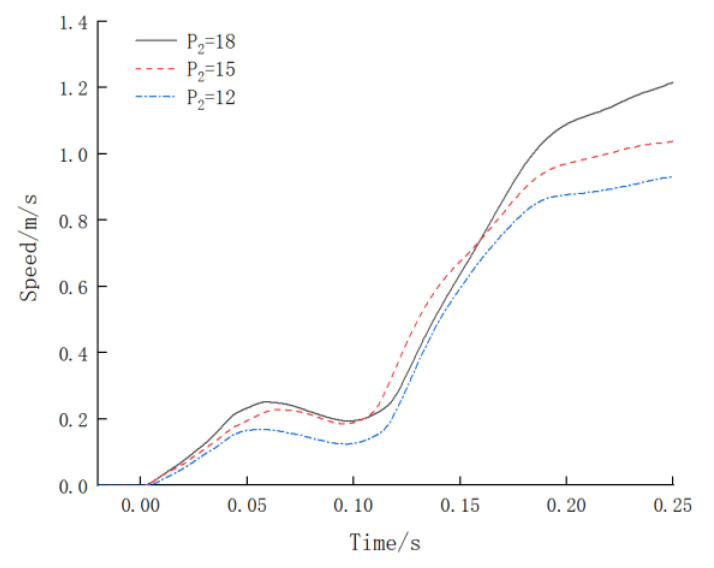
Impact rate–time curve with accumulator working pressure as variable.

**Table 1 materials-16-03892-t001:** Impact test system performance indicators.

Indicators	Parameter
Maximum impact test force	2000 kN
Maximum impact loading rate	1.5 m/s
Test force measurement range	2–100%F.S
Test force indication accuracy	±1%
Main cylinder stroke range	±500 mm
Deformation measurement resolution	0.01 mm
Deformation measurement accuracy	±0.5%F.S
Displacement measuring range	±500 mm
Displacement measurement resolution	0.01 mm
Displacement measurement precision	±1%F.S
Maximum test space	≥3000 mm

**Table 2 materials-16-03892-t002:** Calculation of oil volume for 25 mm diameter steel bars.

Reinforcement Diameter	Antistretching Load	Ultimate Elongation x_max_	Total Hydraulic Pressure Area ΣA	Total Oil Output Q	Minimum Supplied Oil Pressure P
25 mm	450 kN	140 mm	0.0942 m^2^	13.188 L	4.777 MPa

**Table 3 materials-16-03892-t003:** Impact rate control test parameter design.

Test Number	Reinforcement Diameter/mm	Effective Gas Volume *V*_0_/L	Precharge Pressure *P*_0_/MPa	Minimum Working Pressure P_1_/MPa	Maximum Working Pressure P_2_/MPa	Volume Change of Single Tank AccumulatorΔV/L	Total Number of Open Accumulator Tanks N	Total Oil Output Q/L
1	25	63	9	10	12	7.135	6	42.810
2	25	63	9	10	15	14.693	6	88.158
3	25	63	9	10	18	20.034	6	120.203

**Table 4 materials-16-03892-t004:** Accumulator pressure and number required for different loads and velocities.

	Impact Velocity/m·s^−1^	Precharge Pressure *P*_0_/MPa	Minimum Working Pressure *P*_1_/MPa	Maximum Working Pressure *P*_2_/MPa	Number of Open Accumulator Tanks *N*
Φ25 (450 kN)	0.8~1.0	9	10	12	6
	1.0~1.2	9	10	15	6
	1.2~1.4	9	10	18	6
	1.4~1.6	9	10	20	6
Φ32 (700 kN)	0.8~1.0	9	10	12	7
	1.0~1.2	9	10	15	7
	1.2~1.4	9	10	18	7
	1.4~1.6	9	10	20	7
Φ40 (1100 kN)	0.8~1.0	9	10	12	9
	1.0~1.2	9	10	15	9
	1.2~1.4	9	10	18	9
	1.4~1.6	9	10	20	9

## Data Availability

Data available on request from the authors.

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
