# Peer review of "Development of an Instantaneous Loading Impact Test System for Containment of a Nuclear Power Plant during Aircraft Impact on Steel Bar Joints"

_materials, 2023, doi:10.3390/ma16103892_

Round 1

Reviewer 1 Report

Comments: The explanation given regarding the components and their application is clear and understandable. However, the test verification provided in section 4 could be better for developing a new setup. This area must be strengthened. I recommend that the following comments are considered before acceptance.

·       In the abstract, the authors can precisely explain the novelty of this impact test system compared to other standard models available.

·       The title mentions the development of an instantaneous loading impact test system for the containment of nuclear power plants against aircraft impact. The maximum impact rate of the developed impact system is 1.5m/s. Will this realize the real-time conditions?

·       The authors can summarize a few accidents or case studies in a table that necessitates the large-tonnage instantaneous loading impact test system.

·       A typo error in line 143, etc., Please check.

·       Was there any energy loss during oil flow via the cylinder system? Did the authors verify and consider these energy losses?

·       Since this is a low-velocity impact setup, the vibration of the target and the system will significantly influence the results. How were these vibrations controlled?

·       Did the authors try to measure the vibration generated during extreme loading conditions? 

·       The accumulator pressure required for different loads and velocities can be tabulated.

·       In Figure 11, after 0.04 s, the force curve increases after a short plateau region. Why.  

·       Figure 12 shows why the speed increases after the samples encounter a stress plateau (see Fig 11). What does the sudden drop in speed signify?

·       Overall, this investigation is more like a scientific report than a research paper. What's your scientific contribution? Please explain in detail the novelty of this study. Most of the information in section 2 is already available, which can be summarized or moved to support information.

·       The most crucial part of developing a new impact setup is to verify the output result using standard materials. There must be more than the test verification provided in section 4 to verify the machine's efficiency. How did the authors demonstrate the accuracy of the setup provided in Table 1?

Author Response

Dear reviewer, thank you for taking the time to review my article and provide suggestions for improvement. I am honored to address the issues you raised regarding this paper. I have carefully answered each question you posed and modified the relevant content based on your requirements. Please find detailed responses in the attached document. Once again, thank you for your valuable input.

Reviewer 2 Report

Review report: Development of an instantaneous loading impact test system for containment of nuclear power plant against aircraft impact on steel bar joints

1. Discuss the Novelty of the work. Specify the motive behind the work.

2. Shorten the length of the introduction section and add key published work and try to make a bridge between current and previous published work.

3.  Remove the common information presented in the experimental section. Add complete detail about the experimental section.

4. Technical discussion is very poor.

5. Shorten the length of the conclusion section and add key bullet points only.

Author Response

(The authors gave the same response as above.)

Reviewer 3 Report

The authors an instantaneous loading impact test system for containment of nuclear power plant against aircraft impact on steel bar joints. The authors propose a high-tonnage instant loading impact testing system with the 358 maximum impact force of 2000 kN and maximum impact velocity of 1.5 m/s.

The paper seems to be an original contribution to this field however it may be accepted after the minor revision as proposed below.

1-      Literature review is weak. The authors must emphasize and highlight the novelty of the research by providing a more detailed tabular literature review (from the latest and relevant research), elaborating the key parameters and approaches used by various researchers. The authors should review more research articles from 2020-2021-2022-2023.

2-      An Ishikawa diagram must be included to explain the factors effecting the performance of the system.

3-      Test Parameters: Justification of the choice of Impact test system performance indicators in Table 1 must be discussed with references.

4-      Standards: Other than "JJG139-2014 Tensile, Pressure and 285 Universal Testing Machine Calibration Regulations, other relevant standards related to steel structures specifically structural strength requirement for Nuclear Power plants, which consequently define the Impact test system performance indicators in Table 1 must be discussed too.

5-      Future Recommendations: conclusion must also provide more generic benefits and applications of the research carried out, furthermore, the limitations of the research, as well as the future research recommendations must also be discussed in more detail.

Author Response

(The authors gave the same response as above.)

Round 2

Reviewer 2 Report

Accepted